# Design of PG-Surfactants Bearing Polyacrylamide Polymer Chain to Solubilize Membrane Proteins in a Surfactant-Free Buffer

**DOI:** 10.3390/ijms22041524

**Published:** 2021-02-03

**Authors:** Taro Shimamoto, Tatsuki Nakakubo, Tomoyasu Noji, Shuhei Koeda, Keisuke Kawakami, Nobuo Kamiya, Toshihisa Mizuno

**Affiliations:** 1Department of Life Science and Applied Chemistry, Graduate School of Engineering, Nagoya Institute of Technology, Gokiso-cho, Showa-ku, Nagoya, Aichi 466-8555, Japan; t.shimamoto.444@nitech.jp (T.S.); t.nakakubo.266@stn.nitech.ac.jp (T.N.); 2Research Center for Advanced Science & Technology, The University of Tokyo, 4-6-1 Komaba, Meguro-ku, Tokyo 153-8904, Japan; tnoji@protein.rcast.u-tokyo.ac.jp; 3Department of Nanopharmaceutical Sciences, Graduate School of Engineering, Nagoya Institute of Technology, Gokiso-cho Showa-ku, Nagoya, Aichi 466-8555, Japan; s.b.shuhei0316@gmail.com; 4The OCU Advanced Research Institute for Natural Science & Technology (OCARINA), Osaka City University, 3-3-138 Sugimoto-cho, Sumiyoshi, Osaka 558-8585, Japan; keisuke.kawakami@riken.jp (K.K.); rss78535@yahoo.co.jp (N.K.)

**Keywords:** solubilization surfactants, membrane protein, surfactant-free, photosystem I, polyacrylamide

## Abstract

The development of techniques capable of using membrane proteins in a surfactant-free aqueous buffer is an attractive research area, and it should be elucidated for various membrane protein studies. To this end, we examined a method using new solubilization surfactants that do not detach from membrane protein surfaces once bound. The designed solubilization surfactants, DKDKC_12_K-PA*_n_* (*n* = 5, 7, and 18), consist of two parts: one is the lipopeptide-based solubilization surfactant part, DKDKC_12_K, fand the other is the covalently connected linear polyacrylamide (PA) chain with different *M*_w_ values of 5, 7, or 18 kDa. Intermolecular interactions between the PA chains in DKDKC_12_K-PA*_n_* concentrated on the surfaces of membrane proteins via amphiphilic binding of the DKDKC_12_K part to the integral membrane domain was observed. Therefore, DKDKC_12_K-PA*_n_* (*n* = 5, 7, and 18) could maintain a bound state even after removal of the unbound by ultrafiltration or gel-filtration chromatography. We used photosystem I (PSI) from *Thermosynecoccus vulcanus* as a representative to assess the impacts of new surfactants on the solubilized membrane protein structure and functions. Based on the maintenance of unique photophysical properties of PSI, we evaluated the ability of DKDKC_12_K-PA*_n_* (*n* = 5, 7, and 18) as a new solubilization surfactant.

## 1. Introduction

One-third of natural proteins are classified as membrane proteins, and these have crucial roles in various biological events occurring at cell membranes. Photosynthesis is one such membrane-mediated event that participates in different unique membrane proteins. In higher plants and cyanobacteria, photosystem I (PSI) and photosystem II (PSII), integrated in thylakoid membranes, play pivotal roles in generating reductive electrons from water using sunlight [1]. In natural thylakoid membranes, the generated reductive electrons are used for NADPH synthesis via ferredoxin NADP^+^ oxidoreductase (FNR) [2], but exchanging this reductive enzyme to other reductive enzymes or catalysts the artificial photosynthesis systems in vitro and in vivo were recently examined [3]. If the reductive electrons are directly poured into the electrode, it leads to solar cell construction [4]. To combine PSI with reductive enzymes or catalysts, such as molecular metal catalysts [5], Pt nanoparticles [6], or hydrogenases [7] in vitro, direct modification via adsorption or chemical bonds [8] and indirect electrochemical connection via electrodes or metal nanoparticles [9] have been studied. However, the construction of more sophisticated artificial photosynthesis systems [10] consisting of PSI, reductive catalysts, and electron transfer mediators, and sacrificial reductants using solubilization surfactants for membrane proteins to treat PSI in an aqueous buffer could become a decisive hindrance; if PSI could be treated in an aqueous buffer similar to water-soluble proteins, it would allow flexible design of molecular circuits using PSI. The techniques able to treat membrane proteins in an aqueous buffer, identical to water-soluble proteins, such as the construction of various semi-artificial molecular sensors and drug screening samples consisting of different natural membrane proteins is also an interesting research area [11].

Some methods capable of treating membrane proteins in a surfactant-free buffer have been reported so far. Its developments have particular relevance to structural analytical techniques of membrane proteins using cryo-microscopy [12] and multi-dimensional NMR [13]. For structural analysis by cryo-microscopy, sample preparation of membrane proteins with fewer contaminants is indispensable for obtaining high-quality TEM images for structural analysis. Some new surfactants such as lauryl maltose neopentyl glycol [14] and amphipols [15] have a high affinity for membrane proteins; therefore, the unbound surfactants are known to separate without precipitation by dialysis or ultrafiltration. On the other hand, some amphiphilic block copolymers are known to extract membrane proteins directly from bio-membranes [16,17]. By using the deuterated membrane proteins expressed in *E. coli* cell membranes, structural analysis of membrane proteins solubilized with these copolymers using multi-dimensional NMR has been reported [18]. Considering these situations, further developments of new surfactants or techniques capable of treating membrane proteins in surfactant-free buffers are expected. However, developments in these techniques are still limited to date.

Meanwhile, we recently studied Gemini-type peptide surfactants, PG-surfactants [19,20,21,22,23]. The basic molecular scaffold of PG-surfactants consists of three constituents: the linker peptide from to 3–5 residues (X), two alkylamidomethyl-modified Cys residues at both sides of the linker peptide, and the peripheral peptides (Y and Z) at the N- and C-terminal sides of the alkylamidomethyl-modified Cys residues (Figure 1a). From a set of screenings on hydrophilic peptide sequences at X, Y, and Z, we found that two PG-surfactants, DKDKC_12_K and DKDKC_12_D, were able to function as solubilization surfactants for membrane proteins [20]. Furthermore, by the tandem connection of the -Cys(C_12_)-Asp-Lys-Asp-Lys-Cys(C_12_)- units in DKDKC_12_K and DKDKC_12_D via flexible (Gly)_4_ linkers in one molecule, we succeeded in designing the high molecular weight (MW) solubilization surfactants, Bis-D_3_-DKDKC_12_ (MW ~3 kDa), Bis-K_3_-DKDKC_12_ (MW ~3 kDa), and Tris-D_3_-DKDKC_12_ (MW ~4.3 kDa) [21]. Interestingly, an increase in Mw enhanced the affinity between membrane proteins and surfactants, thereby allowing effective solubilization, even for lower concentration ranges (<0.0005 wt%). In this study, by conjugating a hydrophilic polymer unit with DKDKC_12_K, we designed different types of high-Mw surfactants, which can function as a solubilization surfactant for membrane proteins (Figure 1b). We hypothesized that due to the high affinity between polymer chains concentrated on membrane surfaces, these surfactants would not detach once bound to membrane protein surfaces, thereby generating a method to treat membrane proteins in the surfactant-free buffer. In this case, because the attached polymer chains would also interact with membrane protein surfaces, to choose hydrophilic polymers, giving less impact on tertiary structure of membrane proteins, is necessary. Therefore, as a hydrophilic polymer, in this study, we chose poly(acrylamide) (PA). We synthesized PA with different MW (*M*_n_ = 5, 7, and 18 kDa) and characterized the solubilization ability of the conjugates with DKDKC_12_K. DKDKC_12_K-PA*_n_* (*n* = 5, 7, and 18, Figure 1b), and the PA units were introduced at the Y position of DKDKC_12_K.

## 2. Results and Discussion

### 2.1. Design of Solubilization Surfactants Bearing Linear Polyacrylamide (PA) Polymer Chain

To synthesize the polymer-appended DKDKC_12_K, we separately synthesized the hydrophilic polymer and the surfactant parts and then joined them. For the surfactant part, we chose the PG-surfactant DKDKC_12_KC, in which one Cys residue was added to the N-terminal side of DKDKC_12_K (Appendix A). Because DKDKC_12_K has a higher molecular weight (MW 1330) than the general low MW solubilization surfactants such as *n-octyl-β-d-glucopyranoside* (β-OG, MW 292) and *n-dodecyl-β-d-maltopyranoside* (β-DDM, MW 511), it was tolerant of introducing other functional groups to the N- or C-terminal of the same molecule without losing its original solubilization function for membrane proteins [20,23]. We expected that DKDKC_12_K would also conjugate hydrophilic polymers without losing the original solubilization function. An additional Cys residue was used for conjugation with the hydrophilic polymer part.

For the hydrophilic polymer part, we chose polyacrylamide (PA) in this study. As PA is used as a gel material for protein electrophoresis [24] and immobilization substrates for proteins [25], choosing it as a polymer for solubilization surfactants would be reasonable to reduce structural damage to membrane proteins. To examine the impact of *M*_n_ on the solubilization functions of polymer-appended DKDKC_12_K, we synthesized PA with different *M*_n_ values of 5, 7, and 18 kDa by reversible addition-fragmentation chain transfer (RAFT) polymerization [26]. Further, to conjugate with DKDKC_12_KC, the RAFT initiator containing the dithiopyridyl (DTP) group, BSTP pyridyl disulfide [27] was used for the synthesis of PA bearing the DTP groups at the terminal, PA*_n_*-DTP (*n* = 5, 7, and 18, Scheme 1). The synthesized PA*_n_*-DTP*s* were subjected to gel permeation chromatography (GPC), and the calculated *M*_n_*s* and polydispersity indexes (PDI) are listed in Table 1.

Joining of DKDKC_12_KC and PA*_n_*-DTA via S-S bond (Scheme 2) was performed according to Scheme 2, and the target conjugates were purified by reverse-phase high-performance liquid chromatography (RP-HPLC, Appendix A). Compound identification was performed by dividing the PA and DKDKC_12_KC units again via reducing the isolated conjugates using DTT, and RP-HPLC identified each divided unit. Hereafter, the polymer-appended DKDKC_12_K*s* were named DKDKC_12_K-PA*_n_* (*n* = 5, 7, 18).

Before studies on PSI solubilization using new surfactants, we first characterized the fundamental micelle formation properties of DKDKC_12_K-PA*_n_* based on dynamic light scattering (DLS) measurements and critical aggregation concentrations (CAC*s*) (Table 2). With an increase in surfactant concentration, the fluorescence intensity of ANS in a buffer increased with a single inflection point. The plots of *F*_478_ vs. surfactant concentration are shown in Appendix A. Since the concentrations at this inflection point, evaluated from the cross point of double linear-fittings, correspond to the CAC of the amphiphilic molecules, we could determine the CAC*s* (mol/L) of DKDKC_12_K-PA_5_, DKDKC_12_K-PA_7_, and DKDKC_12_K-PA_18_ and they were found to be 51, 46, and 47 µM, respectively; the CACs with mol/L as a unit were calculated from CAC with wt% as a unit by the assumption of MW of DKDKC_12_K-PA_5_, DKDKC_12_K-PA_7_, and DKDKC_12_K-PA_18_, as 5900, 8300, and 19,200, respectively. The modification of PA to DKDKC_12_K hampered micelle formation between the DKDKC_12_K moieties, but the maintenance of micelle formation property was observed for all DKDKC_12_K-PA*_n_*. The DLS profiles of 0.1 wt% DKDKC_12_K-PA_5_, DKDKC_12_K-PA_7_, and DKDKC_12_K-PA_18_ in phosphate buffer gave a single peak at 13, 25, and 29 nm, respectively, and an apparent increase in micelle diameters was observed with the increase in *M*_n_ of the PA unit. These data also support the idea that DKDKC_12_K units in DKDKC_12_K-PA*_n_* could form micelles similar to those of DKDKC_12_K [20] even after the introduction of PA units.

### 2.2. Solubilization of Photosystem I from Thermosynecoccus (T.) Vulcanus by DKDKC_12_K-PA_n_ (n = 5, 7, and 18)

To evaluate the solubilization ability of DKDKC_12_K-PA*_n_* for membrane proteins, in this study, we used PSI from *T. vulcanus* as a representative membrane protein. PSI is a trimeric supramolecular pigment-protein complex (total MW of 1068 kDa); each PSI unit includes 12 protein subunits, 96 chlorophyll a (Chl a) molecules, and 3 [4Fe-4S] clusters [28]. In higher plants and cyanobacteria, it exists in thylakoid membranes and participates in the reductive side light reaction. Because surfactants have unique photophysical properties derived from the hierarchical supramolecular organization of these components, this protein could be used as a probe to evaluate impacts of surfactants on protein structure and functions from impacts on their photophysical properties [20,21]. The PSI samples solubilized with the buffer K2 (40 mM HEPES−NaOH (pH 7.8), 100 mM NaCl, 15 mM CaCl_2_, and 15 mM MgCl_2_) containing 0.1 wt% DKDKC_12_K-PA*_n_* were prepared by the surfactant-exchange method as previously reported [20]. The observed absorption spectra at 298 K and the fluorescence spectra at 77 K are shown in Figure 2. Absorption spectra in the 300–800 nm range mainly originated from 96 molecules of antenna Chl a, coordinated in PsaA and PsaB subunits at PSI integral membrane domain. Therefore, maintenance of the absorption spectrum could strongly suggest maintenance of the tertiary structure, especially at the integral membrane domain. The fluorescence spectrum of PSI in the 650–800 nm range at 77 K corresponds to the formation of the red chlorophyll state by Chl a molecules in the PsaA and PsaB scaffolds. This fluorescence is also a unique characteristic of Chl a molecules in native PSI. Therefore, maintenance of the fluorescence spectrum also supports the maintenance of the tertiary structure at the integral membrane domain of PSI. As shown in Figure 2, all PSI samples, solubilized by 0.1 wt% DKDKC_12_K-PA*_n,_* showed similar spectra using 0.1 wt% β-DDM as a solubilization surfactant (control of the native state), suggesting that all DKDKC_12_K-PA*_n_* showed successful solubilization of PSI without denaturation, similar to the parent PG-surfactants, DKDKC_12_K.

Upon using general low MW solubilization surfactants, membrane proteins are solubilized in a buffer by covering the hydrophobic surfaces of the membrane integral domain with surfactant micelles. Therefore, if the surfactant concentration is less than the critical micelle concentration (CMC), most surfactants cannot form micelles and cannot solubilize membrane proteins in a buffer. However, if the affinity of surfactants to membrane proteins or between surfactants concentrated onto membrane protein surfaces was sufficiently high, the necessary number of surfactant molecules able to solubilize was not governed under surfactant CMC*s*. In short, if the affinity between DKDKC_12_K-PA*_n_* and PSI or intermolecular interactions between the introduced PA chains in DKDKC_12_K-PA*_n_* is high enough, the isolation of the conjugates of PSI and DKDKC_12_K-PA*_n_* was probable. Therefore, we attempted to isolate the conjugates by separation of unbound DKDKC_12_K-PA*_n_* from the PSI sample. Using an ultrafiltration unit (cut-off MW: 100 kDa), we removed the unbound DKDKC_12_K-PA*_n_* from the PSI samples (theoretical concentration of DKDKC_12_K-PA*_n_* after ultrafiltration was less than 0.00001 wt%), prepared in the buffer K2 with 0.1 wt% DKDKC_12_K-PA*_n_*. As surfactant concentrations decreased to less than 0.00001 wt%, the reference PSI sample solubilized with 0.1 wt% β-DDM became insoluble (Figure 3d). However, those with DKDKC_12_K-PA*_n_* maintained good solubility. These samples were further subjected to gel permeation chromatography (GPC) using buffer K2 without including any solubilization surfactants, and the DKDKC_12_K-PA*_n_*-bound PSI*s* were finally obtained as a water-soluble conjugate in a surfactant-free buffer (Figure 3a–c). Upon this surfactant-free condition, the residual concentrations of DKDKC_12_K-PA*_n_s* in PSI samples were quite less than their CMCs. With considering that hydrophobic interaction of DKDKC_12_K-PA*_n_s* to the hydrophobic surfaces of PSI is mainly originated only from two C12 chains in DKDKC_12_K-PA*_n_s*, this stable binding of DKDKC_12_K-PA*_n_s* to PSI molecules could be occurred by effective intermolecular interactions between the PA chains in DKDKC_12_K-PA*_n_s*.

To characterize the conjugated structure and properties of the soluble state of PSI with DKDKC_12_K-PA*_n_*, we first analyzed the photophysical properties by UV-vis and fluorescence spectroscopy (Figure 4). As shown in Figure 4, all conjugates with DKDKC_12_K-PA_5_, DKDKC_12_K-PA_7_, and DKDKC_12_K-PA_18_ showed typical absorption peaks of PSI at 423 and 680 nm in Figure 4a, similar to the case of solubilizing in a buffer with 0.1 wt% β-DDM (control of the native state) [20,21]. On the other hand, an increase in baseline, less than 600 nm region was also observed, which was different from that of solubilizing in a buffer with 0.1 wt% DKDKC_12_K-PA*_n_*, especially for the conjugates with DKDKC_12_K-PA_5_ and DKDKC_12_K-PA_7_ (Figure 2a). This suggests the formation of PSI aggregations. The fluorescence spectral peak at 720 nm, derived from the red-chlorophyll state, was retained for all PSI samples. If the elimination of Chl a from the PSI scaffold occurred, its fluorescence peak should be observed at 680 nm. However, no fluorescent peak at 680 nm was observed, suggesting that PSI maintained the native state in the conjugates.

The light-induced electron transfer activity of PSI is a useful probe to assess them and can be evaluated by concentration change in solubilized oxygen using an oxygen electrode. In the presence of MV^2+^ molecules in a solution, the electrons generated by photoexcitation at the P_700_ special pair first migrate following the potential gradients through phylloquinone and the [4Fe-4S] cluster sites (*F*_x_, *F*_A_, and *F*_B_) and are finally trapped in MV^2+^ molecules in a solution. Because the dissolved oxygen immediately quenches the one-electron reductant of MV^2+^, the initial oxygen consumption rate can be considered identical to the initial rate of light-induced electron transfer in PSI [20,29]. After reducing the resultant hole at the special pair (P700^+•^) by sodium ascorbate assisted by dichloroindophenol (DCIP), one sequential electron transfer process initiated by light irradiation is completed. If protein denaturation occurred, the passage of electron migration was also damaged. As a result, the electron transfer rate could be reduced. The electron transfer rates per single PSI unit were calculated from the background-subtracted oxygen consumption data, and the obtained initial rates are summarized in Table 3. Although covering the PSI surface with PA units might hamper electron transfer to the MV^2+^ and DCIP in a buffer, comparable electron transfer rates of those in a buffer with 0.1 wt% β-DDM (control of the native state). This data also supported that PSI in the PSI/DKDKC_12_K-PA*_n_* conjugates maintain native characteristics, suggesting the maintenance of the tertiary structure of the membrane integral and the extracellular domains.

In order to examine the morphologies of the conjugates of PSI and DKDKC_12_K-PA*_n_*, we performed TEM observations. Each TEM sample was prepared by short adsorption of the conjugates onto the polyvinyl alcohol (PVA) layer-coated TEM grids and staining with sodium phosphotungstate. In the case of conjugates with DKDKC_12_K-PA_5_ and DKDKC_12_K-PA_7_, the formation of plate-type 2D aggregations (less than 100 nm of wide and ~7 nm thickness), orienting horizontally (blue enclosing regions and arrows) or perpendicularly (red enclosing regions and arrows) to the surface of the PVA layer was observed, as shown in the left and center low TEM images of Figure 5. On the other hand, the conjugates with DKDKC_12_K-PA_18_ showed homogeneously dispersed spherical morphologies, having ~20 nm diameter (purple enclosing regions and arrows) in the right low TEM images of Figure 5. From X-ray structural analysis, PSI is known to have a low columnar structure and a diameter of about 20 nm [29]. The reference TEM image of the PSI sample, prepared from the PSI solution in a buffer with 0.1 wt% β-DDM, gave circular morphologies with a diameter of ~20 nm (data not shown). This meant that the circular morphologies in the right low TEM images of Figure 5 correspond to the conjugate PSI portion. Upon solubilizing with DKDKC_12_K-PA_18_, DKDKC_12_K-PA_18_ molecules bind the integral membrane domain of PSI and locate around PSI molecules. Therefore, the bound DKDKC_12_K-PA_18_ was expected to be located between the circular ~20 nm morphologies with a lower height than the PSI.

Based on the morphologies and rigorous assessments of the photophysical properties, we found that DKDKC_12_K-PA_18_ is the best PA-appended PG-surfactant among the other DKDKC_12_K-PA*_n_* studied, solubilizing PSI in a surfactant-free buffer without inducing PSI aggregation and denaturation.

## 3. Materials and Methods

### 3.1. Materials

Rink-amide AM resin (200–400 mesh) was purchased from Merck Biosciences (Darmstadt, Germany). N-(9-fluorenylmethoxycarbonyl) (Fmoc)-protected-amino acids, 1-hydroxybenzotriazole (HOBT), 2-(1H-benzotriazole-1-yl)-1,1,3,3-tetramethyluronium hexafluorophosphate (HBTU), N,N-diisopropylethylamine (DIEA), piperidine, trifluoroacetic acid (TFA), and N-methyl pyrrolidone (NMP) were purchased from Watanabe Chemical Industries (Hiroshima, Japan). Acrylamide and VA-057 were purchased from Wako Pure Chemical Industries, Ltd. (Osaka, Japan). Poly(ethylene glycol) average *M*_n_ = 1500 g/mol (PEG1500), *n*-dodecyl-β-d-maltopyranoside (β-DDM), dichloroindophenol (DCIP), *L*(+)-Ascorbic Acid Sodium Salt, and methyl viologen dichloride dihydrate (MV^2+^) were purchased from Sigma-Aldrich (St. Louis, MO, USA). The RAFT initiator containing the dithiopyridyl (DTP) group, BSTP pyridyl disulfide, was synthesized according to previous study [27]. PSI derived from *T. vulcanus* was prepared similar to the previous study [21]. Unless otherwise stated, other chemicals and reagents were obtained commercially and used without further purification. BSTP disulfide was synthesized as following the previous study.

### 3.2. Synthesis of PG-Surfactant, DKDKC_12_KC

PG-surfactant, DKDKC_12_KC, was synthesized on a Rink-amide AM resin using commercially available Fmoc-protected amino acids and our synthesized Fmoc-Cys(C_12_)-OH [19]. For condensation onto the resin, standard coupling reagents (HOBT/HBTU/DIEA) were used. The N-terminus of PG-surfactants was end-capped with Ac_2_O. After cleavage of the synthesized PG-surfactants from the resin using TFA/H_2_O (95/5), the crude PG-surfactants were purified by reversed-phase high-performance liquid chromatography (RP-HPLC) with a core-shell-type ODS column (Kinetex, Shimadzu, Japan). A linear-gradient of CH_3_CN and H_2_O, both including 0.1 vol% TFA, was utilized as eluent (Appendix A). Product identification was checked by high-resolution ESI-TOF (electrospray ionization time-of-flight) mass spectroscopy (Appendix A).

DKDKC_12_KC: HRMS (ESI-TOF, [M + H]^+^): calcd. for C_68_H_128_N_15_O_15_S_2_, 1433.8298; found, 1433.8282.

### 3.3. Synthesis of Polyacrylamide Having Dithiopyridyl Terminal Group (PA_n_-DTP, n = 5, 7, 18) by RAFT Polymerization

PA*_n_*-DTP (*n* = 5, 7, 18, *M*_n_ = 5, 7, 18 kDa), having 2-pyridyldisulfide groups at the terminal, were synthesized by the RAFT technique with molar ratio of monomer/initiator/BSTP pyridyl disulfide [27] of 1124/1/2 for PA (18 kDa), 562/1/2 for PA (7 kDa), and 281/1/2 for PA (5 kDa). For instance, PA (18 kDa) was synthesized as follows. AM (2.00 g, 28.1 mmol), VA-057 (10.4 mg, 25.0 µmol), and BSTP pyridyl disulfide (22.1 mg, 50.0 µmol) were dissolved in DMSO/H_2_O (5:3, 16 mL) in a round-bottom flask and sealed with a septum. The flask was degassed by freeze thawing with nitrogen and subsequently placed in a preheated hot-bath at 50 °C for 105 min. The resulting polymer was precipitated 3 times from acetone and dried in vacuo. Yield: 1.41 g (71%). The *M*_n_ and polydispersity index (PDI) of the synthesized polymers were estimated by a gel permeation chromatography (GPC), respectively. These estimated data of *M*_n_ and PDI were summarized in Table 1.

### 3.4. Synthesis of the PA(5, 7, 18 kDa)-Appended DKDKC_12_KC, DKDKC_12_K-PA_n_ (n = 5, 7, 18)

To a N_2_-substituted water (3 mL), PA with dithiopyridyl group (*M*_n_ = 5, 7, 18 kDa, 7 mmol) and DKDKC_12_KC (3.0 mg, 3.5 µmol) was added and it was stirred for 2 h. at ambient temperature. It was purified by RP-HPLC with a core-shell-type ODS column (Kinetex, Shimadzu, Japan). A linear-gradient of CH_3_CN and H_2_O, both including 0.1 vol% TFA, was utilized as eluent. The HPLC profiles of DKDKC_12_K-PA*_n_* (*n* = 5, 7, 18) were summarized in Appendix A.

### 3.5. Dynamic Light Scattering Measurements of PG-Surfactant Assemblies

The concentrations of each DKDKC_12_K-PA*_n_* (*n* = 5, 7, 18) in 100 mM phosphate buffer (pH 7) were set at 0.1 wt% and the mean hydrodynamic diameters of PG-surfactant assemblies for each concentration at 25 °C were estimated using a Zetasizer Nano ZS (Malvern Instruments, Ltd., Malvern, UK).

### 3.6. Critical Aggregation Concentration (CAC) Determination for PG-Surfactants Using 8-Anilino-Naphtharene-1-Sulfonic Acid (ANS)

The CACs of DKDKC_12_K-PA*_n_* (*n* = 5, 7, 18) in 100 mM phosphate buffer (pH 7) were evaluated by the fluorescent method [30] similar to the previous studies [20,21,23].

### 3.7. Replacement of Solubilization Surfactant via PEG Precipitation

The PSI sample, solubilized with DKDKC_12_K-PA*_n_* (*n* = 5, 7, 18) in the buffer K2 [40 mM HEPES–NaOH (pH 7.8), 100 mM NaCl, 15 mM CaCl_2_, and 15 mM MgCl_2_], containing 0.1 wt% of DKDKC_12_K-PA*_n_* (*n* = 5, 7, 18) was prepared similar to the previous studies [20,21].

### 3.8. Isolation of the Conjugates of PSI and DKDKC_12_K-PA_n_ (n = 5, 7, 18)

The unbound-DKDKC_12_K-PA*_n_*s (*n* = 5, 7, 18) were first removed by ultrafiltration (Amicon Ultra 0.5 mL Centrifugal Filters, Devise NMWL 100 kDa) for the PSI sample, solubilized with DKDKC_12_K-PA*_n_* (*n* = 5, 7, 18) in the buffer K2 (40 mM HEPES–NaOH (pH 7.8), 100 mM NaCl, 15 mM CaCl_2_, and 15 mM MgCl_2_). Then these were subjected to gel permeation chromatography (1 cm (i.d.) × 30 cm (h), Superose 6, GE Healthcare, Milwaukee, WI, USA) was applied using the buffer K2 as an eluent.

### 3.9. Evaluation of Photo-Induced Electron-Transfer Rate of PSI Based on Decreases in O_2_ Concentration

Measurements of the O_2_ uptake rate to evaluate photo-induced initial electron transfer rate of PSI were conducted at 25 °C using a Clark-type O_2_ electrode (Hansatech Instruments, DW1, Oxygen Electrode Unit; Norfolk, VA, USA), similar to the previous study [20,21].

### 3.10. Fluorescence Spectrum of PSI at 77 K

The fluorescence spectrum of the conjugates of PSI and DKDKC_12_K-PA*_n_* (*n* = 5, 7, 18) in the buffer K2 was observed similar to the previous study [20,21].

### 3.11. TEM Measurements of the Conjugates of PSI and DKDKC_12_K-PA_n_ (n = 5, 7, 18)

TEM images were obtained with JEM-z2500 (JEOL). All the samples were prepared by dry-cast of protein solutions supported on a poval-coated Cu grid (400 mesh, Okenshoji Co., Ltd., Tokyo, Japan). Membrane protein concentration was 5 µg Chl*a*/mL of the isolated PSI conjugates with DKDKC_12_K-PA*_n_* (*n* = 5, 7, 18). This solution was dropped on the Cu grid and left to stand for approximately 5 min at room temperature. Droplets were removed with filter paper and stained by sodium phosphotungstate solution (2 wt%) in pure water. This sample was rinse three times with pure water.

## 4. Conclusions

In this study, we constructed PA-modified PG-surfactants, DKDKC_12_K-PA*_n_*, and examined its ability to solubilize membrane proteins. All surfactants could solubilize PSI in a buffer containing 0.1 wt% of each surfactant without any denaturation. However, under surfactant-free conditions (i.e., after removal of the unbound surfactants from the sample solution), the ability to solubilize PSI in a buffer was different; DKDKC_12_K-PA_5_ and DKDKC_12_K-PA_7_ could solubilize PSI in a surfactant-free buffer without protein denaturation, but several PSI aggregations were observed. In contrast, DKDKC_12_K-PA_18_ could solubilize PSI without inducing protein denaturation and aggregation. This should be reasoned by the difference in affinity of PA chains, concentrated onto PSI surfaces; PA chains with 18 kDa of Mn had enough affinity without inducing elimination of DKDKC_12_K-PA_18_ molecules from the PSI surface.

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
