# Peer review of "Design of PG-Surfactants Bearing Polyacrylamide Polymer Chain to Solubilize Membrane Proteins in a Surfactant-Free Buffer"

_ijms, 2021, doi:10.3390/ijms22041524_

Round 1

Reviewer 1 Report

The work described by Toshihisa Mizuno and co-workers involves the synthesis of lipopeptides covalently connected to linear polyacrylamide. Authors have carefully studied the chemical coupling of the hybrid conjugates as well as their ability to stabilize the PSI protein system (Photosystem I from Thermosynecoccus vulcanus). I would recommend the publication of this article in International Journal of Molecular Sciences after taking into account the following comments

Introduction: The authors describe too much in detail the structural elements related to their previous work on lipopeptides when these elements could be summarized in 2 or 3 sentences. On the other hand, the motivation for coupling those lipopeptides to a polymer is not sufficiently presented and it would be interesting to provide some bibliographical references on the subject. The choice of the polymer also deserves to be argued.

TYPOS: line 110: Mn = 1500 (units are missing); line 120: condensation should be replaced by coupling; line 178: What is the percentage of phosphotungstic acid in solution for TEM? line 203: the following sentence: "The Mns of the synthesized PAn-DTP were analyzed by GPC..." sounds inappropriate. A Mn is never analyzed by GPC... line 216: the conditions must be added to the scheme (arrow) and the legend.

Discussions: the proposed conjugates combining polymer and lipopeptides provide higher CAC values (table 2). This observation contradicts the fact that these molecules stabilize protein systems better than polymer-free lipopeptides. It would certainly be necessary to show that the CAC in the presence of the proteins is lower than the lipopeptide alone to validate the use of the compound. Alternatively, it would be important to add the results obtained with lipopeptides alone (figure 2 for example). On the other hand, could the electron transfer activity allow an evaluation of the quantity of encapsulated PSI system (in mass %?).

Author Response

  1. The authors describe too much in detail the structural elements related to their previous work on lipopeptides when these elements could be summarized in 2 or 3 sentences.

As following the referee’s comment, we shortened the description of previous works on lipopeptides in the revised manuscript (Line 83, page 2).

  1. On the other hand, the motivation for coupling those lipopeptides to a polymer is not sufficiently presented and it would be interesting to provide some bibliographical references on the subject. The choice of the polymer also deserves to be argued.

As following the referee’s comment, we added the following sentences (italic part) about discussion on choice of polymer in the revised manuscript (Line 94-97, page 3, yellow-highlighted).

In this case, because the attached polymer chains would also interact with membrane protein surfaces, to choose hydrophilic polymers, giving less impacts on tertiary structure of membrane proteins, is necessary. Therefore, as a hydrophilic polymer, in this study, we chose poly(acrylamide) (PA). We synthesized PA with different MW (Mn = 5, 7, and 18 kDa) •••••••.

  1. line 110: Mn = 1500 (units are missing)

As following the referee’s comment, we added the unit “g/mol” in the revised manuscript (Line 108, page 3, yellow-highlighted).

  1. line 120: condensation should be replaced by coupling.

As following the referee’s comment, we exchanged “condensation” to “coupling” in the revised manuscript (Line 118, page 3, yellow-highlighted).

  1. line 178: What is the percentage of phosphotungstic acid in solution for TEM?

As following the referee’s comment, we added concentration of sodium phosphotungstate in pure water in the revised manuscript (Line 175-176, page 5, yellow-highlighted).

•• stained by sodium phosphotungstate solution (2 wt%) in pure water.

  1. line 203: the following sentence: "The Mns of the synthesized PAn-DTP were analyzed by GPC..." sounds inappropriate. A Mn is never analyzed by GPC...

We modified description about GPC analysis as following in the revised manuscript (Line 201, page 5, yellow-highlighted).

The synthesized PAn-DTPs were subjected to gel permeation chromatography (GPC) •••••.

  1. line 216: the conditions must be added to the scheme (arrow) and the legend.

As following the referee’s comment, we added “under N2 atmosphere” above the scheme (arrow) in the revised manuscript (Line 213, page 6).

  1. The proposed conjugates combining polymer and lipopeptides provide higher CAC values (table 2). This observation contradicts the fact that these molecules stabilize protein systems better than polymer-free lipopeptides. It would certainly be necessary to show that the CAC in the presence of the proteins is lower than the lipopeptide alone to validate the use of the compound. Alternatively, it would be important to add the results obtained with lipopeptides alone (figure 2 for example). On the other hand, could the electron transfer activity allow an evaluation of the quantity of encapsulated PSI system (in mass %?).

In general, if molecular weight (MW) of solubilization surfactants was increased, critical micelle concentration (CMC) of surfactants could also be increased. It is because steric hinderance necessary to assemble surfactant molecules in micelle morphologies could be increased, according to increase in molecular size of surfactants. But from our previous study on the high MW solubilization surfactants Bis-D3-DKDKC12 and Tris-D3-DKDKC12, due to large surface area and particle size of membrane proteins such as PSI, effective concentrations of surfactants enable to solubilize PSI in aqueous buffer were less than their CMCs. It meant that molecular-packing of surfactant molecules upon covering the hydrophobic surface of membrane proteins (i.e. to solubilize membrane proteins in an aqueous buffer) is different from that on forming spherical micelle morphologies by themselves. We expected that similar phenomenon should be occurred for DKDKC12K-PAn; effective concentrations of DKDKC12K-PAn enable to solubilize PSI in aqueous buffer were less than their CMCs. Actually, due to the intermolecular interactions between the PA chains in DKDKC12K-PAn onto PSI surfaces, DKDKC12K-PAn could maintain the bound-state, even after removement of unbound ones. In order to clarify these points, we added the following phrase and sentences in the revised manuscript (L277, page 7 and L288-292, page 8, yellow-highlighted).

<L277, page 7>

In short, if the affinity between DKDKC12K-PAn and PSI or intermolecular interactions between the introduced PA chains in DKDKC12K-PAn is high enough, ••••••.

< L288-292, page 8>

Upon this surfactant-free condition, the residual concentrations of DKDKC12K-PAns in PSI samples were quite less than their CMCs. With considering that hydrophobic interaction of DKDKC12K-PAns to the hydrophobic surfaces of PSI is mainly originated only from two C12 chains in DKDKC12K-PAns, this stable binding of DKDKC12K-PAn to PSI molecules could be occurred by effective intermolecular interactions between the PA chains in DKDKC12K-PAn.

Reviewer 2 Report

The manuscript "Design of PG-surfactants bearing polyacrylamide polymer chain to solubilize membrane proteins in a surfactant-free buffer" by Shimamoto T. et al. reported about the sytnthesis and application of new gemini-type peptide surfactans characterized by linear polyacrilamide chains with different molecular weights (5, 7, 18 KDa) that acts as spacer covalently linked at both sides to two lipopeptide-based solubilization surfactants DKDKC12K, already published in previous paper by the same research group (Langmuir 2013, 29, 11667-11680, ref 21).

The authors studied the extraction and solubilization of membrane proteins with these new PG-surfactans. In particular, the photosystem I (PSI) from thylakoid membranes of Thermosinecoccus Vulcanus was used as a model to test the efficiency of the PG-surfactans with respect to standards one beta-DDM in buffer aqueous solution.

The work seems convincing and consistent with previous papers, and the use of a polymeric linker introduces an interesting novelty compared to the literature. Moreover, the authors presented also the morphological characterization of the conjugates of PSI and DKDKC12K-PA obtained wih TEM too (Figure 5 pag 11) that should of high interest.

Unfortunately, I recommend major revisions of this work as TEM images are used to draw the coclusions, but they show very low quality. As a matter of fact, Figure 5d (the reference) is blurry and it is not clean what they are observing. The other images are also unclear, even for stained samples.

I suggest to the authors for the sake of the readers to improve the quality of the images and to highlight with arrows of different colours what they are comparing in the different images with respect to the reference one to better support their conclusions.

Author Response

  1. Unfortunately, I recommend major revisions of this work as TEM images are used to draw the conclusions, but they show very low quality. As a matter of fact, Figure 5d (the reference) is blurry and it is not clean what they are observing. The other images are also unclear, even for stained samples.I suggest to the authors for the sake of the readers to improve the quality of the images and to highlight with arrows of different colours what they are comparing in the different images with respect to the reference one to better support their conclusions.

Actually, it is tough to get high-resolution TEM images of the PSI/DKDKC12K-PAn conjugates due to technical limitations. But in order to help understanding of the expected molecular structures of the aggregates observed in TEM images of the PSI/DKDKC12K-PAn conjugates, we added the schematic illustration of the expected molecular structures of the aggregates and the representative ones to be noticed in TEM images were indicated by red, blue, and purple enclosing regions and arrows in the revised Figure 5. According to modifications of Figure 5, we also modified the figure caption as following (yellow-highlighted).

<Figure caption of Figure 5>

TEM images of the isolated PSI/DKDKC12K-PAn conjugates (PSI/DKDKC12K-PA5 conjugates (left low), PSI/DKDKC12K-PA7 conjugates (center low), and PSI/DKDKC12K-PA18 conjugates (right low)) and the schematic illustrations of the expected molecular structures of aggregates, observed in TEM images (upper). 2D aggregates of PSI particles solubilized with DKDKC12K-PA5 or DKDKC12K-PA7 are indicated by red or blue enclosing regions and arrows. While single PSI particles solubilized with DKDKC12K-PA18 are indicated by purple enclosing regions and arrows. Accelerate voltage, 200 kV.

Furthermore, in order to help understanding of the revised Figure 5, we modified several sentences in the revised manuscript (L342-351, page 9-10, yellow-highlighted) as following.

< L342-351, page 9-10>

In the case of conjugates with DKDKC12K-PA5 and DKDKC12K-PA7, the formation of plate-type 2D aggregations (less than 100 nm of wide and ~ 7 nm thickness), orienting horizontally (blue enclosing regions and arrows) or perpendicularly (red enclosing regions and arrows) to the surface of the PVA layer was observed, as shown in the left and center low TEM images of Figure 5. Whereas, the conjugates with DKDKC12K-PA18 showed homogeneously dispersed spherical morphologies, having ~20 nm diameter (purple enclosing regions and arrows) in the right low TEM images of Figure 5. From X-ray structural analysis, PSI is known to have a low columnar structure and a diameter of about 20 nm [29]. The reference TEM image of the PSI sample, prepared from the PSI solution in a buffer with 0.1 wt% β-DDM, gave circular morphologies with a diameter of ~20 nm (data not shown). This meant that the circular morphologies in the right low TEM images of Figure 5 correspond to the conjugate PSI portion.

Round 2

Reviewer 1 Report

This revised manuscript basically addressed all my previous concerns. The authors responded to all comments and adressed the required changes to their manuscript. In this context, I would recommend that this article be accepted for publication.

Reviewer 2 Report

In the revision of the manuscript "Design of PG-surfactants bearing polyacrylamide polymer chain to solubilize membrane proteins in a surfactant-free buffer" by Shimamoto T. et al., the authors aswered to the questions of this reviewer and, even if they didn't provide new TEM images they improve the clarity of their discussion of the results.